# Identifying CO_2_ Seeps in a Long-Dormant Volcanic Area Using Uncrewed Aerial Vehicle-Based Infrared Thermometry: A Qualitative Study

**DOI:** 10.3390/s22072719

**Published:** 2022-04-01

**Authors:** Dan Mircea Tămaș, Boglárka Mercédesz Kis, Alexandra Tămaș, Roland Szalay

**Affiliations:** 1Department of Geology and Center for Integrated Geological Studies, Babeș-Bolyai University, 400084 Cluj-Napoca, Romania; boglarka.kis@ubbcluj.ro (B.M.K.); alexandra.tamas1@ubbcluj.ro (A.T.); szalay.j.roland@gmail.com (R.S.); 2MTA-ELTE Volcanology Research Group, ELTE University, 1053 Budapest, Hungary

**Keywords:** thermal images, UAV, CO_2_, gas emission, Romania

## Abstract

Ciomadul is a long-dormant volcanic area in the Eastern Carpathians of Romania. The study site, the Stinky Cave, and the surrounding areas are well-known for CO_2,_ and H_2_S seeps. The gases from these seeps come with high flux and are of magmatic origin, associated with the volcanic activity of Ciomadul. In this study, an Uncrewed Aerial Vehicle coupled with a thermal infrared sensor is used to identify new seeps. In order to achieve this, we carried out several field campaigns, coupling image acquisition with the creation of digital outcrop models and orthomosaics. The study was carried out at low ambient temperatures to identify strong thermal anomalies from the gasses. Using this qualitative study method, we identified several new seeps. The total emission of the greenhouse gas CO_2_ in the Ciomadul area and other similar sites is highly underestimated. The practical application of this method will serve as a guide for a future regional rollout of the thermal infrared mapping and identification of CO_2_ seeps in the area.

## 1. Introduction

In recent years, Infrared Thermometry (IRT) has seen several advances and widespread usage across the earth sciences, these applications ranging from monitoring of temperature variations, heat fluxes, the volatile activity of volcanic and/or geothermal areas [1,2,3,4,5,6] to the identification of different lithologies through their emissivity [7] and quantification of rockfall hazards [8].

Quiescent but often reawakening volcanic and geothermal sites could represent a potential hazard. Hence, they require continuous surveillance that can be best provided through remote sensing, using satellite thermal images combined with other geophysical and geochemical techniques [9,10,11,12,13]. Techniques for the remote surveillance of volcanic and geothermal areas have their benefits as they provide data even in hazardous or inaccessible regions, while ground-based thermal sensors can be installed at a relatively safe place on a volcanic crater rim [1,2,3,4]. However, recent advances in Uncrewed Aerial Vehicles (UAV) sensors, such as Thermal Infrared (TIR) sensors, enable the monitoring of large areas at high resolution. Their increasing use is mainly due to their versatility and low cost [6]. Using UAVs, several quiescent and active volcanic and geothermal areas have been surveyed recently, establishing them as a powerful tool in real-time data gathering and offering significant contribution to hazard assessment and risk management [5,6,14,15,16,17]. Small drones equipped with different chemical sensors could provide new approaches and research opportunities in air pollution and emission monitoring and study atmospheric trace gases [18,19,20] or even detect hazardous volatile leaks [18].

Structures from Motion (SfM) and UAV-based SfM have been increasingly used to create 3D outcrop models or Digital Outcrop Models (DOM) and high-resolution Digital Elevation Models (DEM) [19,20,21]. UAV photography can be used to create 3D outcrops, as well as aid in mapping, providing images for inaccessible areas, and the creation of orthorectified aerial photographs [19,21,22].

Ciomadul is a long-dormant volcanic area (Romanian Eastern Carpathians), where several attempts have been made to quantify the total CO_2_ emitted from this volcano and understand the origin of the gas emissions [23,24,25,26]. The gas emissions from Ciomadul and neighboring areas appear in different forms: (i) focused free gas emissions, (ii) dissolved gases of CO_2_ and H_2_S-rich mineral water springs, and (iii) areas where diffuse degassing is present. Previously, the identification of these CO_2_ and H_2_S-rich gas seeps/focused emissions and mineral water springs has mainly been through local knowledge. Historical and analytical information has been made available related to the chemical composition of some mineral springs and mofettes [27,28]. However, these do not account for all springs and seeps, therefore having a method of identifying and monitoring gas emissions is especially important in the case of the dormant Ciomadul volcano, where a significant output of CO_2_ has recently been quantified [25]. Additionally, inspection of the area in order to identify, and map gas emissions is also problematic due to the high bear population in the area.

In this study, UAV-based IRT and SfM techniques were applied to identify gas emission areas (hotspots). The confirmation of gas emissions and their composition has been determined using measurements performed with a Multi-GAS instrument. This study aims to demonstrate the efficiency of the qualitative identification method and best practices.

## 2. Geological Overview

Ciomadul Volcano is located at the southeastern edge of the Carpathian-Pannonian Region, at the southern end of the Călimani-Gurghiu-Harghita volcanic chain [29,30,31,32,33] (Figure 1). It is part of a post-collisional volcanic belt composed of andesitic to dacitic volcanoes, developed as part of the Carpathian orogen system [34,35,36]. The volcano complex intruded and developed on the Early Cretaceous clastic flysch sedimentary unit of the Eastern Carpathians consisting of an alternation of sandstones, calcareous sandstones, limestones, and clays/marls, and having a thickness up to 2500 m [34,37,38,39].

The Ciomadul volcanic complex comprises several lava domes truncated by two explosion craters called Mohos and Saint Anna [32,33]. These are surrounded by further isolated lava domes (Bába Laposa, Haramul Mic, Dealul Mare, Büdös-Puturosul and Bálványos, Figure 1) [40]. Volcanism at the Ciomadul volcanic dome field started around 1 Ma. The lava domes of Büdös-Puturosul were formed 642 ± 44 kyrs ago, while the Bálványos dome was formed 583 ± 30 kyrs ago [40,41]. From a petrological point of view, the eruptive products consist mostly of dacites [31,32].

The study area is located around the Stinky Cave of Turia village, at the Büdös/Puturosul mountain/volcanic dome, part of the Ciomadul volcanic area (Figure 1). Several caves are found here and are associated with CO_2_ and H_2_S gas emissions. These gases are of magmatic origin, associated with the volcanic activity of Ciomadul [23,24,26]. The Stinky Cave (46°7′11.28″ N, 25°56′54.51″ E) opens on the southeast side of the Büdös/Puturosul volcanic dome at an altitude of 1052 m. The cave is an abandoned sulfur mine from which sulfur was extracted to produce gunpowder. The gas level is associated with sulfur deposition, with the resulting sulfur crystals being 3–4 mm. The yield of cave gas was first determined as 734,000 m^3^ of CO_2_ and 2850 m^3^ of H_2_S per year [41]. More recent gas yield determination has been estimated at 1923 tons of CO_2_ per year [25]. Comparing the results with the values obtained by [42], which is converted into 1413 tons/year, we find that the values are similar, so the gas yield in the cave is relatively constant.

## 3. Materials and Methods

### 3.1. UAV Data Acquisition

#### 3.1.1. Technical Specifications of the UAV Cameras

Aerial photography was acquired using a DJI Mavic 2 Enterprise Dual UAV. It is equipped with a 12 MP visual camera (RGB) with a 1/2.3″ CMOS sensor. The visual camera has a lens with a field of view of ~85° and a 24 mm (35 mm format equivalent) lens with an aperture of f/2.8. The drone is also equipped with an Integrated Radiometric FLIR^®^ Thermal Sensor. It is an Uncooled VOx Microbolometer with a horizontal field of view of 57° and an f/1.1 aperture. The sensor resolution is 160 × 120 (640 × 480 image size) and has a spectral band of 8–14 μm.

#### 3.1.2. Hardware Limitations

The DJI Mavic 2 Enterprise Dual has several limitations that have been considered and accounted for during the data acquisition for this study. Because of these limitations, the UAV thermal data has allowed for a qualitative study rather than a quantitative one.

First of all, the UAV TIR camera does not export radiometric data. The exported thermal image is only a raster that has a fixed color scheme and range that the user pre-sets before image acquisition. No further changes are possible after image acquisition. The spectral data that can be derived and analyzed from many TIR sensors is also not available with this TIR camera. As a result, flow rates could not be calculated. Identifying different gases was also not possible with the thermal sensor-equipped by the DJI Mavic 2 Enterprise Dual.

#### 3.1.3. RGB and Infrared Thermometry Survey

Temperature measurements inside the cave during the fieldwork are ~6–8 °C even when the outside temperature is colder (i.e., −3 °C). For this reason, four different surveys were carried out during wintertime on two separate days (Table 1). This ensured that the temperature after sunset was below freezing, thus ensuring suitable conditions to measure the warm thermal signatures associated with the gas emissions.

The images were acquired from 10 cm to 120 m above the lift-off ground level (1073 m). Before the acquisition, four ground control points (GCP) were put in place, and their locations were measured. Only manual photograph acquisition was used due to the vertical faces of the outcrop, the mountain’s steepness, and high vegetation density to avoid unwanted collisions. In order to ensure a good correlation (alignment) between the images, they were acquired to provide approximately 80% front overlap and 70% side overlap. 

To account for the hardware limitations (of the thermal camera) mentioned above, fixed temperature color scales have been used (see Table 1). Two different thermal palettes have been used for the image acquisition (HotMetal and Rainbow; see Table 1). The different thermal palettes were used to identify which would work best with the SfM software.

Multi-Spectral Dynamic Imaging (FLIR MSX™) was used to enhance the correlation (alignment) between the acquired TIR images. This option adds visible light details to the thermal image without diluting the image. In order to ensure the correct operation of the FLIR MSX™ feature during the dark, the Mavic 2 Enterprise Spotlight (with 2400 lumens and 17° FOV) has been used. The FLIR MSX™ feature did not work during the low-light image acquisition from a higher altitude (>40 m).

### 3.2. 3D SfM Models

To create the DOM, DEM, and Orthorectified models, we used Agisoft Metashape Professional (v.1.8.2) [43]. The first step in processing the data was aligning the photographs. The dense point cloud was generated after the alignment was performed and the region of interest selected. Point confidence was used to filter out the highly uncertain points. From the resulting dense point cloud, the DEM, textured mesh, and orthomosaics were generated. The position match between the UAV-based orthomosaic and satellite imagery is almost exact; thus, there is high confidence in the orientation of the DOM as well.

The workflow used for creating the IRT SfM models is very much similar to the one used for making the RGB models (see [22] for more details). One of the most significant differences is that although the FLIR MSX™ feature added visible light details to the thermal image, it still was not enough to ensure correct alignment between all the photographs. For this reason, ten other markers were introduced in key places in the area of interest to enhance the image matching process. 

### 3.3. Measuring Gas Emissions

The CO_2_, CH_4_, and H_2_S compositions of the gas vents were realized using a portable, Multi-component Gas Analyser System (Multi-GAS), an instrument designed and used worldwide for volcanic gas monitoring [44,45]. Our instrument is specially designed for low-temperature gas emissions and provides real-time data on the CO_2_ (%), CH_4_ (%), and H_2_S (ppm) concentrations of the free gas emissions. The Multi-GAS was equipped with two Gascard II IR type spectrometers to determine CO_2_ (0 to 100 % range), CH_4_ (0 up to 7% range), and one electrochemical H_2_S sensor that measured the concentrations between 0 to 200 ppm.

The gas collected from the different vents was pumped to the detectors that communicated through a data logger with a smartphone. The real-time measurement could be followed until the concentrations of the different gas species reached stable values.

## 4. Results

### 4.1. Infrared Thermometry Surveys

#### 4.1.1. IRT Survey 1

This survey was made during daylight (see Table 1) when the studied outcrop and cave entrance were partially exposed to direct sunlight. The main scope of this survey was to acquire the RGB images for the successful construction of the RGB digital outcrop model. Still, the TIR images recorded warm thermal anomalies in the Stinky Cave (Figure 2b,c). The maximum recorded radiometric temperatures in the warm part of the cave interior were similar to those recorded in the area where the warmer gases melted the snow (Figure 2). Most of the warm thermal anomalies observed outside the cave (Figure 2b,c) could also be caused by exposure to sunlight and, as a contrast to the snow cover (Figure 2a,c).

#### 4.1.2. IRT Survey 2

The second IRT survey was acquired after sunset when the ambient temperature fell below freezing (see Table 1), and the cliff face, vegetation, and exposed soil temperatures were also lower. Three significant warm thermal anomalies were identified during this survey (Figure 3). One of them was previously known (the Stinky Cave; Figure 3b,c,f), but two other significant warm thermal anomalies were discovered (Figure 3d,e). The identified thermal anomalies presented elongated shapes (Figure 3) and were visible even from 70 m (above the take-off altitude). The use of the spotlight during the acquisition aided us in distinguishing visible features on the thermal images (with FLIR MSX™).

Along with the more significant warm thermal anomalies (Figure 3), on the images acquired from a higher altitude (approximately 70 m), we also identified several areas/patches with warm thermal anomalies, but not as significant as the main three.

#### 4.1.3. IRT Surveys 3 and 4

Both surveys in the second campaign of IRT were carried out after sunset. During IRT survey 4 the ambient temperature and that of the cliff face and vegetation were slightly cooler compared with IRT survey 3 (0.5–1 °C). The main difference between the two surveys is the thermal palette used for acquisition (HotMetal for IRT survey 3 and Rainbow for IRT survey 4; Table 1, Figure 4). The scope of the change in the thermal palette was to evaluate which of them would provide better SfM model results.

During this campaign, several other (smaller) hotspots have been identified. The three main ones were still providing strong warm thermal anomalies. As in the previous campaign, the identified thermal anomalies extended downdip of the hotspot locations (Figure 4).

### 4.2. Structure from Motion Models

The RGB SfM photogrammetric models (Figure 5b) were successfully used to identify structural features that may promote the seeps’ localization and provide a tie point for the TIR SfM models (Figure 5a). The TIR SfM models (Figure 5a) and the orthomosaics offer a clear and integrated view of all the warm thermal anomalies. We identified several smaller anomalies in the models created based on IRT surveys 3 and 4. The three most significant hotspots identified (one of them being the Stinky Cave) present elongated warm thermal anomalies that extend downdip or laterally (Figure 5a). The two main newly identified hotspots have been cross-correlated between the TIR and RGB SfM models. They originate from areas in the outcrop that are intensively fractured (Figure 2a,c and Figure 5b). Fracture/fault orientations have been measured on the RGB models and show two main trends of ~65/355 and ~80/60 (Figure 5b).

### 4.3. Gas Measurements

After the UAV-based mapping campaigns, the most significant positive thermal anomalies and some of the minor ones have been measured. The measured IRT hotspots were confirmed as gas emissions (mainly CO_2_; see measured compositions in Table 2). Prior to the IRT surveys, the only previously identified gas seep was the Stinky Cave. Inside the cave, the high flux gas emission is CO_2_-rich, see Table 2. According to the gas composition, the gas mix is denser than the air, and therefore it flows outside like a river at the bottom of the cave entrance and creates a gas corridor in front of the cave which can be detected, on thermal images (see Figure 2b,c, Figure 3b,c,f, Figure 4a,c and Figure 5a). In addition to the data collected in the cave, the previously unidentified dry gas vents also became visible due to the observed thermal hotspots. CO_2_ dominated these observed dry gas vents, with values ranging from 97.46 to 78.65%. There were noticeable native sulfur precipitations in several cases that indicated high H_2_S presence, the concentrations at the measurement points varied between 127.6 and 166.31 ppm. In all measurements, the CH_4_ concentration was similar to the Stinky Cave and showed a high composition range compared to the geological context, varying between 4.08 and 3.06 % (see Table 2).

## 5. Discussion

### 5.1. Thermal Anomalies in the IRT Surveys and SfM Models

The use of drones combined with lightweight thermal cameras and/or miniaturized devices for the measurements of concentrations of different gas species has proven to be a powerful tool in the study of volcanic and geothermal areas [6,46,47]. 

The warm thermal anomalies identified using the UAV-based IRT surveys and TIR SfM models have been measured in the field using the Multi-GAS. We thus confirmed that all the identified thermal anomalies represent gas emissions, mainly CO_2_. The elongated warm thermal anomalies identified extending downdip of the Stinky Cave and the other two main hotspots are interpreted as areas where the gases emitted in those seep areas were flowing downdip (some of them being heavier than air; Figure 2, Figure 3, Figure 4 and Figure 5).

Warm air temperature and warmer background temperatures highly influenced the results of the IRT for this type of survey (see Figure 2). Similar temperatures between the background and the gas seeps affected the identification of these anomalies. An even more significant masking effect (overprint) was created when the studied area was exposed to sunlight (Figure 2).

Successful IRT work has been carried out on outcrops during summertime at times with no sun exposure of the outcrop but had other areas of research focus [8]. In the case of this study, summertime testing was carried out in order to investigate if the gas seeps provide identifiable cold thermal anomalies. This approach did not yield positive results as there was no effect of the active gas seeps (Table 2) on the TIR images.

### 5.2. Implications for the Study of Gas Emissions 

The surroundings of our study area the Büdös/Puturosul volcanic dome are characterized by CO_2_-rich focused emissions and diffuse degassing. Secondary minerals like native sulfur and alum depositions already suggested some signs of the presence of gas emissions. No direct vents were found with the naked eye except in the Stinky Cave, where the interface of the atmosphere and the mofettic gas is visible due to the native sulfur deposits on the cave wall (Figure 1).

Diffuse degassing structures were previously identified at other sites within the Büdös/Puturosul volcanic dome, where a significant amount of CO_2_ output, more than 5000 tonnes/year, was measured and calculated [25]. When groundwater gets in contact with these vents, the gases reach the surface bubbling and they are easier to identify. The diffuse degassing structures are not always easy to find, especially when no direct signs of the vents are visible. This study highlights the potential for applying UAV-based methods to identify dry gas vents that have not been identified or mapped.

A combination of UAVs and TIR cameras has been successfully applied at several sites, for example at Le Salinelle, Italy, the thermal manifestations of a mud volcano were mapped [14]; at Geysir geothermal field, Iceland, a detailed thermal anomaly map was conducted, and from the thermal mosaic thousands of distinct anomalies suggesting individual vents were identified [16]. A similar combined method was used for the imaging of the Lusi mud eruption, Indonesia, a harsh environment that is inaccessible for direct observations [5].

The temperatures of the gasses in the study area (between 6–8 °C) are not as high as in other geothermal areas, still, the UAV could provide information on the thermal properties of the site. Relying on previous observations, it was essential to choose the right environmental circumstances in which the temperature of the gas emissions could be identified as they are a relatively low-level anomaly compared to the ambient daytime temperature. Using the TIR images, hotspots with high CO_2_ concentrations were identified (Table 2) that are impossible to detect with the naked eye. The pattern of the vents revealed by the thermal images suggests that the degassing system of the Stinky Cave could be more significant than previously accounted for. The similarity in CO_2_ concentration suggests that the smaller gas vents are likely to be connected with the cave through faults and fractures, which act as a pathway for the gases.

The identification of the smaller vents suggested by the thermal images was confirmed using our Multi-GAS instrument, which demonstrated high CO_2_ concentrations of the small vents, similar to the Stinky Cave (Table 2). Thus, the method proved to be suitable for the detection of small structures within a low-temperature degassing area and could be of key importance in the future mapping of similar low-temperature sites. 

Moreover, after or even during the identification of the spots, compositional and flux measurements can be taken to better quantify the CO_2_ output of the area and to reveal possible structural patterns of the gas emissions. The methodology is already well established in the case of active volcanoes such as the Turrialba and Masaya, Central America, where miniaturized UAV-mounted instrumentation, also including a small MultiGAS was used for the compositional measurement of volcanic gases and the obtained values were in good agreement with ground-based measurements [47].

### 5.3. Way Forward

A larger-scale survey should be set up to map a large area like the Ciomadul volcanic area from a significant height. The three main hotspots and other warm patches have been identified from heights of +100 m when the air temperature was −3 °C. Future extensive area surveys will be conducted at or slightly below the minimum operating temperature of the drone (−10 °C). Miniaturized instrumentation may also be tested in the case of the gas emissions in our study area.

This study lays the groundwork for more detailed quantification of the total gas emissions generated in the area of the Ciomadul long-dormant volcano and other such areas in the Romanian Carpathians. These findings will aid others in identifying further examples of such structures, especially with the growing interest in better understanding and quantifying natural and manmade CH_4_ and CO_2_ emissions worldwide.

## 6. Conclusions

This study highlights the ability of the Uncrewed Aerial Vehicle-based thermal infrared imaging technique in identifying discrete CO_2_ emissions in dormant volcanic systems. Using this low altitude airborne monitoring method, new seep locations were identified and measured using a Multi-GAS portable instrument, which is capable of quantifying different gas species on-site and in real-time. The Structure from Motion photogrammetric models provided a clearer view of the seeps and enabled us to identify their precise location. Detailed UAV-based mapping of these areas can lead to a better understanding and quantification of the presence of CO_2_ emissions. The total emission of the greenhouse gas CO_2_ in the Ciomadul area and other similar areas is likely to be highly underestimated. The effective application of this method will serve as a guide for a future regional rollout of the thermal infrared mapping and identification of CO_2_ emissions in the area.

## Figures and Tables

**Figure 1 sensors-22-02719-f001:**
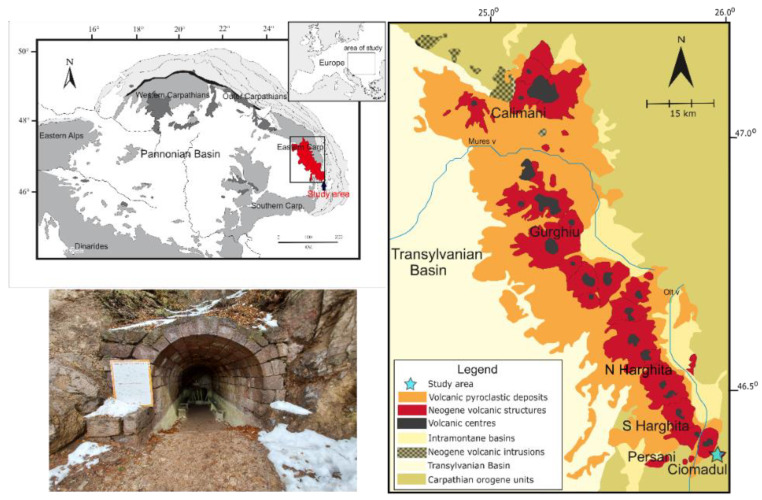
Regional overview of the Carpathian-Pannonian domain (after [29]) with the location of the simplified geological map of the Eastern Carpathians volcanic chain (modified after [30]) highlighted in red. The Ciomadul area is also indicated on with a blue star. Photograph of the Stinky Cave entrance.

**Figure 2 sensors-22-02719-f002:**
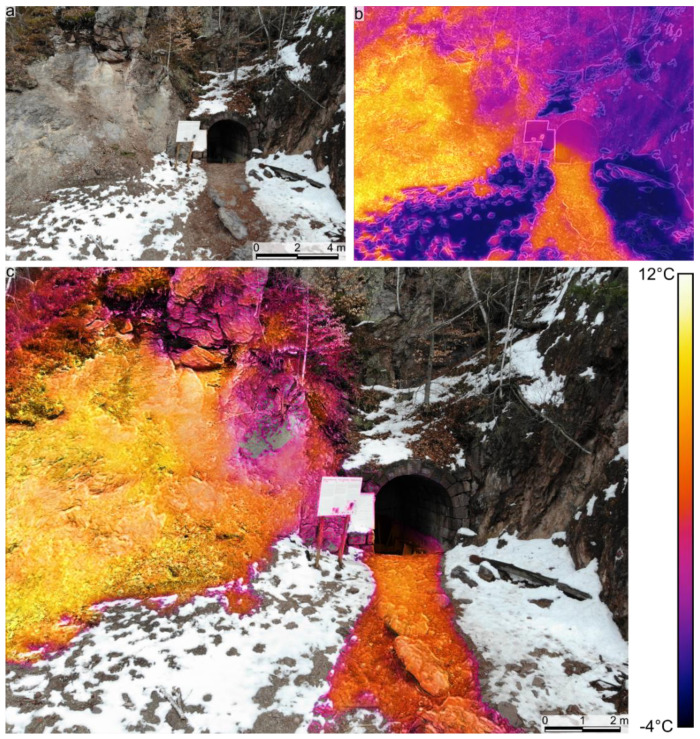
RGB (**a**) and IRT (**b**) photographs of the area of interest (survey 1, Table 1). (**c**) Blend between IRT hotspots and RGB images. Note the high temperatures on the exposed outcrop.

**Figure 3 sensors-22-02719-f003:**
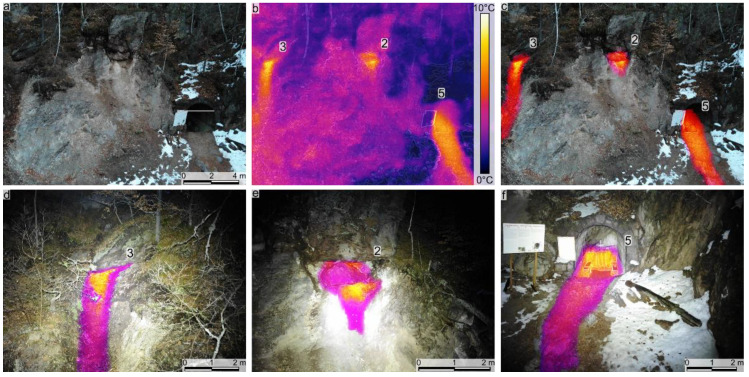
RGB (**a**) and IRT (**b**) photographs of the area of interest (survey 2, Table 1). (**c**–**f**) Blended images between IRT hotspots and RGB photographs. Note that numbers on figures (**b**–**f**) indicate the gas measurement ID (see Table 2).

**Figure 4 sensors-22-02719-f004:**
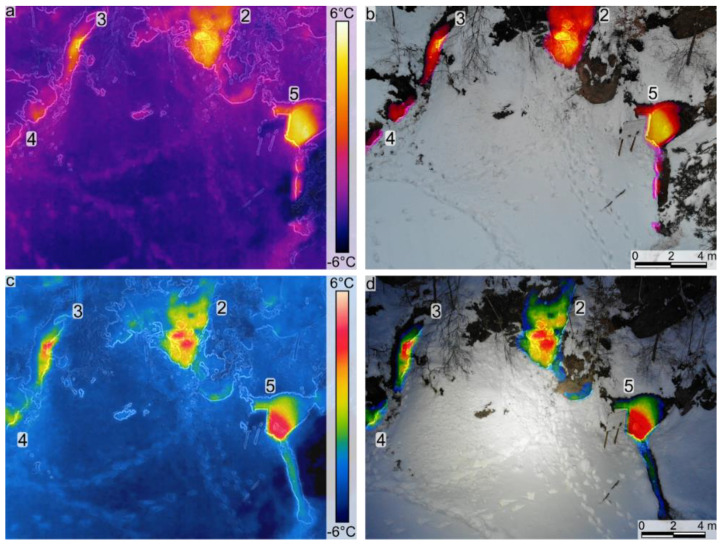
Infrared thermometry photographs (**a**,**c**) and blended images between IRT hotspots and RGB photographs (**b**,**d**). Note the details on the Rainbow thermal palette (**c**,**d**; survey 4, Table 1) in comparison to the HotMetal palette (**a**,**b**; survey 3, Table 1). Note that the numbers indicate the gas measurement ID (see Table 2).

**Figure 5 sensors-22-02719-f005:**
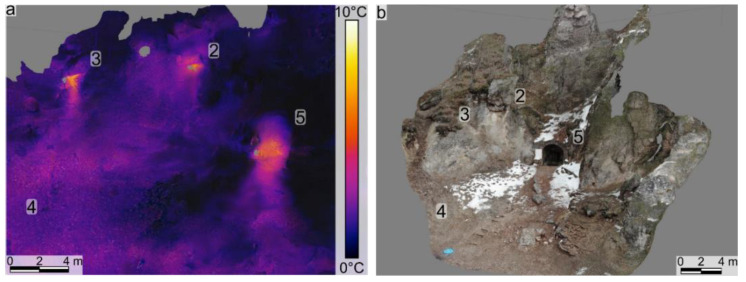
Infrared thermometry (**a**: survey 2) and RGB (**b**: survey 1) structure from motion 3D digital outcrop models of the study area. Note that numbers on figures indicate the gas measurement ID (see Table 2).

**Table 1 sensors-22-02719-t001:** List of Infrared Thermometry (IRT) surveys and details regarding ambient temperatures, thermal palettes, temperature scales, and the number of photographs.

Campaign No.	IRT Survey No.	Acquisition Time (GMT + 2)	Air Temp. (°C)	Thermal Palette	Temp. Scale (°C)	No. of Photographs
1	1	13:00	6	HotMetal	−4–12	329
1	2	18:30	−2	HotMetal	0–10	181
2	3	18:30	−3	HotMetal	−6–6	360
2	4	19:00	−4	Rainbow	−6–6	289

**Table 2 sensors-22-02719-t002:** List of dry gas measurements (see locations in Figure 3, Figure 4 and Figure 5).

ID	Site	Date	Longitude	Latitude	H_2_S (ppm)	CH_4_(%)	CO_2_(%)
1	Dry gas emission 1	21 March 2021	25.948388	46.119762	166.32	4.08	94.57
		23 August 2021			164.84	3.67	97.8
2	Dry gas emission 2	21 March 2021	25.94855	46.11986	164.92	3.85	97.46
3	Dry gas emission 3	21 March 2021	25.94846	46.11982	127.6	3.42	93.66
		23 August 2021			67.14	3.05	91.25
4	Dry gas emission 4	21 March 2021	25.94843	46.11978	165.26	3.06	78.65
		23 August 2021			164.83	3.64	95.79
5	Stinky Cave	21 March 2021	25.94867	46.11980	164.91	3.62	96.63
		23 August 2021			164.89	3.65	97.59

## Data Availability

Not applicable.

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
