# Peer review of "Identifying CO2 Seeps in a Long-Dormant Volcanic Area Using Uncrewed Aerial Vehicle-Based Infrared Thermometry: A Qualitative Study"

_sensors, 2022, doi:10.3390/s22072719_

Round 1

Reviewer 1 Report

The manuscript presents Identifying CO2 seeps in a long-dormant volcanic area using 2 Uncrewed Aerial Vehicle-based Infrared Thermometry: a quali-3 tative study

In general, I appreciate the work that has been done and presented here. The topic is of interest to improve UAS studies, but in my view, the study must be improved for reasons below.

  1. The introductory background is too lengthy that the focus and contributions of this study is only at the end of the chapter.
  2. It needs significant grammatical improvement. I highly recommend having a more proficient English-speaker do a thorough review of the paper.
  3. Many general descriptions on the methodology in section 3 also feel distracting and confusing. In my consideration authors must rewrite the chapter more clearly
  4. The discussion and conclusion section is, in my opinion, much too short. I think that there needs to be a much greater discussion of what was learned from the surveys.  I think that several additional plots, which compare UAV measurement data and structural data; UAV data and degassing data would significantly strengthen the interpretations of this paper.

  1. I suggest to merge chapter 2 with 2.1. Line 102 change H2S with H2S
  2. Line 180: why the title ‘3D SfM models’ if you mentioned the gas composition? Please change the title
  3. Line 261: change table 1 with table 2. In any case I think that the table is incomplete (where are details regarding ambient temperatures?)
  4. Line 271. see figs. 2b, c, 3b, c, f, 4a, c. I suggest to repeat the number and letters (i.e. 2b, 2c…4a,4c…)

Other minor suggestions are in the text but in any case the authors need to improve their work, mention background information only when they are necessary or at least relevant for this study, and also rewrite most of the paper to clearly state the motivation and contribution of this study. I think the paper can be publishable with these significant changes.

Reviewer 2 Report

The Manuscript entitled: "Identifying CO2 seeps in a long-dormant volcanic area using Uncrewed Aerial Vehicle-based Infrared Thermometry: a qualitative study" is very interesting scientific work suitable to sensors journal. The introduction is well organised and fulfill the requirements of a good scientific introduction. 

  • However the Reviewer recommends to not use bucklet of references without deeper explanation. e.g. line 29 and 37 "[1-6] etc." It would be better to separate them and present for example problem a [1], problem b [2], problem c [3] etc.

The methodology is sufficient for this study and other researchers who would like to compare their studies can do this following the description of this methodology presented in this Manuscript.

The results are in accordance to the literature however there is slight lack of organization in section 4. Some things are described in one section and the figure is put in another. Therefore it is a little bit difficult to analyse these results if someone would like to do this on ones own. 

  • It must be reorganized e.g. figure 3 and description gathered together.

In Discussion and Results part the only thing it missing is the comparison to other works.  The Authors have presented sufficient literature survey in the introduction however the conclusion and discussion section have not followed this idea. 

  • The results of this research should be compared with other scientific papers.

Overall merit of the Manuscript is very positive therefore it should be publish after the revision.

Round 2

Reviewer 1 Report

Authors completed the requested revisions and this study can be published in the present form

Reviewer 2 Report

i accept the Authors answers